# Generalizing to Unseen Domains
# via Adversarial Data Augmentation

**Riccardo Volpi**[*,†]
Istituto Italiano di Tecnologia

**Hongseok Namkoong**[*]
Stanford University

**Ozan Sener**
Intel Labs

**John Duchi**
Stanford University

**Vittorio Murino**
Istituto Italiano di Tecnologia
Università di Verona

**Silvio Savarese**
Stanford University

## Abstract

We are concerned with learning models that generalize well to different *unseen* domains. We consider a worst-case formulation over data distributions that are near the source domain in the feature space. Only using training data from a single source distribution, we propose an iterative procedure that augments the dataset with examples from a fictitious target domain that is "hard" under the current model. We show that our iterative scheme is an adaptive data augmentation method where we append adversarial examples at each iteration. For softmax losses, we show that our method is a data-dependent regularization scheme that behaves differently from classical regularizers that regularize towards zero (*e.g.*, ridge or lasso). On digit recognition and semantic segmentation tasks, our method learns models improve performance across a range of a priori unknown target domains.

## 1 Introduction

In many modern applications of machine learning, we wish to learn a system that can perform uniformly well across multiple populations. Due to high costs of data acquisition, however, it is often the case that datasets consist of a limited number of population sources. Standard models that perform well when evaluated on the validation dataset—usually collected from the same population as the training dataset—often perform poorly on populations different from that of the training data [15, 3, 1, 32, 38]. In this paper, we are concerned with generalizing to populations different from the training distribution, in settings where we have no access to any data from the unknown target distributions. For example, consider a module for self-driving cars that needs to generalize well across weather conditions and city environments unexplored during training.

A number of authors have proposed domain adaptation methods (for example, see [9, 39, 36, 26, 40]) in settings where a fully labeled source dataset and an unlabeled (or partially labeled) set of examples from fixed target distributions are available. Although such algorithms can successfully learn models that perform well on known target distributions, the assumption of a priori fixed target distributions can be restrictive in practical scenarios. For example, consider a semantic segmentation algorithm used by a robot: every task, robot, environment and camera configuration will result in a different target distribution, and these diverse scenarios can be identified only after the model is trained and deployed, making it difficult to collect samples from them.

In this work, we develop methods that can learn to better *generalize* to new unknown domains. We consider the restrictive setting where training data only comes from a single source domain. Inspired

---

[*]Equal contribution.
[†]Work done while author was a Visiting Student Researcher at Stanford University.

by recent developments in distributionally robust optimization and adversarial training [34, 20, 12], we consider the following worst-case problem around the (training) source distribution $P_0$

$$\underset{\theta \in \Theta}{\text{minimize}} \quad \underset{P:D(P,P_0)\leq\rho}{\sup} \mathbb{E}_P[\ell(\theta;(X,Y))]. \tag{1}$$

Here, $\theta \in \Theta$ is the model, $(X,Y) \in \mathcal{X} \times \mathcal{Y}$ is a source data point with its labeling, $\ell : \mathcal{X} \times \mathcal{Y} \to \mathbb{R}$ is the loss function, and $D(P,Q)$ is a distance metric on the space of probability distributions.

The solution to worst-case problem (1) guarantees good performance against data distributions that are distance $\rho$ away from the source domain $P_0$. To allow data distributions that have different support to that of the source $P_0$, we use Wasserstein distances as our metric $D$. Our distance will be defined on the semantic space [3], so that target populations $P$ satisfying $D(P,P_0) \leq \rho$ represent realistic covariate shifts that preserve the same semantic representation of the source (*e.g.*, adding color to a greyscale image). In this regard, we expect the solution to the worst-case problem (1)—the model that we wish to learn—to have favorable performance across covariate shifts in the semantic space.

We propose an iterative procedure that aims to solve the problem (1) for a small value of $\rho$ at a time, and does stochastic gradient updates to the model $\theta$ with respect to these fictitious worst-case target distributions (Section 2). Each iteration of our method uses small values of $\rho$, and we provide a number of theoretical interpretations of our method. First, we show that our iterative algorithm is an adaptive data augmentation method where we add adversarially perturbed samples—at the current model—to the dataset (Section 3). More precisely, our adversarially generated samples roughly correspond to *Tikhonov regularized Newton-steps* [21, 25] on the loss in the semantic space. Further, we show that for softmax losses, each iteration of our method can be thought of as a data-dependent regularization scheme where we regularize towards the parameter vector corresponding to the true label, instead of regularizing towards zero like classical regularizers such as ridge or lasso.

From a practical viewpoint, a key difficulty in applying the worst-case formulation (1) is that the magnitude of the covariate shift $\rho$ is a priori unknown. We propose to learn an ensemble of models that correspond to different distances $\rho$. In other words, our iterative method generates a collection of datasets, each corresponding to a different inter-dataset distance level $\rho$, and we learn a model for each of them. At test time, we use a heuristic method to choose an appropriate model from the ensemble.

We test our approaches on a simple digit recognition task, and a more realistic semantic segmentation task across different seasons and weather conditions. In both settings, we observe that our method allows to learn models that improve performance across a priori unknown target distributions that have varying distance from the original source domain.

**Related work**

The literature on adversarial training [10, 34, 20, 12] is closely related to our work, since the main goal is to devise training procedures that learn models robust to fluctuations in the input. Departing from imperceptible attacks considered in adversarial training, we aim to learn models that are resistant to larger perturbations, namely out-of-distribution samples. Sinha et al. [34] proposes a principled adversarial training procedure, where new images that maximize some risk are generated and the model parameters are optimized with respect to those adversarial images. Being devised for defense against *imperceptible* adversarial attacks, the new images are learned with a loss that penalizes differences between the original and the new ones. In this work, we rely on a minimax game similar to the one proposed by Sinha et al. [34], but we impose the constraint in the semantic space, in order to allow our adversarial samples from a fictitious distribution to be different at the pixel level, while sharing the same semantics.

There is a substantial body of work on *domain adaptation* [15, 3, 32, 9, 39, 36, 26, 40], which aims to better generalize to a priori *fixed* target domains whose labels are unknown at training time. This setup is different from ours in that these algorithms require access to samples from the target distribution during training. *Domain generalization* methods [28, 22, 27, 33, 24] that propose different ways to better generalize to unknown domains are also related to our work. These algorithms require

the training samples to be drawn from different domains (while having access to the domain labels during training), not a single source, a limitation that our method does not have. In this sense, one could interpret our problem setting as *unsupervised* domain generalization. Tobin et al. [37] proposes *domain randomization*, which applies to simulated data and creates a variety of random renderings with the simulator, hoping that the real world will be interpreted as one of them. Our goal is the same, since we aim at obtaining data distributions more similar to the real world ones, but we accomplish it by actually *learning* new data points, and thus making our approach applicable to any data source and without the need of a simulator.

Hendrycks and Gimpel [13] suggest that a good empirical way to detect whether a test sample is out-of-distribution for a given model is to evaluate the statistics of the softmax outputs. We adapt this idea in our setting, learning ensemble of models trained with our method and choosing at test time the model with the greatest maximum softmax value.

## 2 Method

The worst-case formulation (1) over domains around the source $P_0$ hinges on the notion of distance $D(P, P_0)$, that characterizes the set of unknown populations we wish to generalize to. Conventional notions of Wasserstein distance used for adversarial training [34] are defined with respect to the original input space $\mathcal{X}$, which for images corresponds to raw pixels. Since our goal is to consider fictitious target distributions corresponding to realistic covariate shifts, we define our distance on the semantic space. Before properly defining our setup, we first give a few notations. Letting $p$ the dimension of output of the last hidden layer, we denote $\theta = (\theta_c, \theta_f)$ where $\theta_c \in \mathbb{R}^{p \times m}$ is the set of weights of the final layer, and $\theta_f$ is the rest of the weights of the network. We denote by $g(\theta_f; x)$ the output of the embedding layer of our neural network. For example, in the classification setting, $m$ is the number of classes and we consider the softmax loss

$$\ell(\theta; (x, y)) := -\log \frac{\exp\left(\theta_{c,y}^\top g(\theta_f; x)\right)}{\sum_{j=1}^m \exp\left(\theta_{c,j}^\top g(\theta_f; x)\right)} \tag{2}$$

where $\theta_{c,j}$ is the $j$-th column of the classification layer weights $\theta_c \in \mathbb{R}^{p \times m}$.

**Wasserstein distance on the semantic space**  On the space $\mathbb{R}^p \times \mathcal{Y}$, consider the following transportation cost $c$—cost of moving mass from $(z, y)$ to $(z', y')$

$$c((z, y), (z', y')) := \frac{1}{2} \|z - z'\|_2^2 + \infty \cdot \mathbf{1}\{y \neq y'\}.$$

The transportation cost takes value $\infty$ for data points with different labels, since we are only interested in perturbation to the marginal distribution of $Z$. We now define our notion of distance on the semantic space. For inputs coming from the original space $\mathcal{X} \times \mathcal{Y}$, we consider the transportation cost $c_\theta$ defined with respect to the output of the last hidden layer

$$c_\theta((x, y), (x', y')) := c((g(\theta_f; x), y), (g(\theta_f; x'), y'))$$

so that $c_\theta$ measures distance with respect to the feature mapping $g(\theta_f; x)$. For probability measures $P$ and $Q$ both supported on $\mathcal{X} \times \mathcal{Y}$, let $\Pi(P, Q)$ denote their couplings, meaning measures $M$ with $M(A, \mathcal{X} \times \mathcal{Y}) = P(A)$ and $M(\mathcal{X} \times \mathcal{Y}, A) = Q(A)$. Then, we define our notion of distance by

$$D_\theta(P, Q) := \inf_{M \in \Pi(P, Q)} \mathbb{E}_M[c_\theta((X, Y), (X', Y'))]. \tag{3}$$

Armed with this notion of distance on the semantic space, we now consider a variant of the worst-case problem (1) where we replace the distance with $D_\theta$ (3), our adaptive notion of distance defined on the semantic space

$$\underset{\theta \in \Theta}{\text{minimize}} \ \sup_P \left\{ \mathbb{E}_P[\ell(\theta; (X, Y))] : D_\theta(P, P_0) \leq \rho \right\}.$$

Computationally, the above supremum over probability distributions is intractable. Hence, we consider the following Lagrangian relaxation with penalty parameter $\gamma$

$$\underset{\theta \in \Theta}{\text{minimize}} \ \sup_P \left\{ \mathbb{E}_P[\ell(\theta; (X, Y))] - \gamma D_\theta(P, P_0) \right\}. \tag{4}$$

---

**Algorithm 1** Adversarial Data Augmentation

---

  **Input:** original dataset $\{X_i, Y_i\}_{i=1,\ldots,n}$ and initialized weights $\theta_0$
  **Output:** learned weights $\theta$

1: **Initialize:** $\theta \leftarrow \theta_0$
2: **for** $k = 1, \ldots, K$ **do**                                      ▷ Run the minimax procedure $K$ times
3:     **for** $t = 1, \ldots, T_{\min}$ **do**
4:         Sample $(X_t, Y_t)$ uniformly from dataset
5:         $\theta \leftarrow \theta - \alpha \nabla_\theta \ell(\theta; (X_t, Y_t))$
6:     Sample $\{X_i, Y_i\}_{i=1,\ldots,n}$ uniformly from the dataset
7:     **for** $i = 1, \ldots, n$ **do**
8:         $X_i^k \leftarrow X_i$
9:         **for** $t = 1, \ldots, T_{\max}$ **do**
10:            $X_i^k \leftarrow X_i^k + \eta \nabla_x \left\{ \ell(\theta; (X_i^k, Y_i)) - \gamma c_\theta((X_i^k, Y_i), (X_i, Y_i)) \right\}$
11:        Append $(X_i^k, Y_i^k)$ to dataset
12: **for** $t = 1, \ldots, T$ **do**
13:     Sample $(X, Y)$ uniformly from dataset
14:     $\theta \leftarrow \theta - \alpha \nabla_\theta \ell(\theta; (X, Y))$

---

Taking the dual reformulation of the penalty relaxation (4), we can obtain an efficient solution procedure. The following result is a minor adaptation of [2, Theorem 1]; to ease notation, let us define the robust surrogate loss

$$\phi_\gamma(\theta; (x_0, y_0)) := \sup_{x \in \mathcal{X}} \left\{ \ell(\theta; (x, y_0)) - \gamma c_\theta((x, y_0), (x_0, y_0)) \right\}. \tag{5}$$

**Lemma 1.** *Let* $\ell : \Theta \times (\mathcal{X} \times \mathcal{Y}) \to \mathbb{R}$ *be continuous. For any distribution $Q$ and any $\gamma \geq 0$, we have*

$$\sup_P \left\{ \mathbb{E}_P[\ell(\theta; (X, Y))] - \gamma D_\theta(P, Q) \right\} = \mathbb{E}_Q[\phi_\gamma(\theta; (X, Y))]. \tag{6}$$

In order to solve the penalty problem (4), we can now perform stochastic gradient descent procedures on the robust surrogate loss $\phi_\gamma$. Under suitable conditions [5], we have

$$\nabla_\theta \phi_\gamma(\theta; (x_0, y_0)) = \nabla_\theta \ell(\theta; (x_\gamma^\star, y_0)), \tag{7}$$

where $x_\gamma^\star = \arg\max_{x \in \mathcal{X}} \left\{ \ell(\theta; (x, y_0)) - \gamma c_\theta((x, y_0), (x_0, y_0)) \right\}$ is an adversarial perturbation of $x_0$ at the current model $\theta$. Hence, computing gradients of the robust surrogate $\phi_\gamma$ requires solving the maximization problem (5). Below, we consider an (heuristic) iterative procedure that iteratively performs stochastic gradient steps on the robust surrogate $\phi_\gamma$.

**Iterative Procedure**    We propose an iterative training procedure where two phases are alternated: a *maximization* phase where new data points are learned by computing the inner maximization problem (5) and a *minimization* phase, where the model parameters are updated according to stochastic gradients of the loss evaluated on the adversarial examples generated from the maximization phase. The latter step is equivalent to stochastic gradient steps on the robust surrogate loss $\phi_\gamma$, which motivates its name. The main idea here is to iteratively learn "hard" data points from fictitious target distributions, while preserving the semantic features of the original data points.

Concretely, in the $k$-th *maximization* phase, we compute $n$ adversarially perturbed samples at the current model $\theta \in \Theta$

$$X_i^k \in \arg\max_{x \in \mathcal{X}} \left\{ \ell(\theta; (x, Y_i)) - \gamma c_\theta((x, Y_i), (X_i^{k-1}, Y_i)) \right\} \tag{8}$$

where $X_i^0$ are the original samples from the source distribution $P_0$. The *minimization phase* then performs repeated stochastic gradient steps on the augmented dataset $\{X_i^k, Y_i\}_{0 \leq k \leq K, 1 \leq i \leq n}$. The maximization phase (8) can be efficiently computed for smooth losses if $x \mapsto c_{\theta^{k-1}}((x, Y_i), (X_i^{k-1}, Y_i))$ is strongly convex [34, Theorem 2]; for example, this is provably true for any linear network. In practice, we use gradient ascent steps to solve for worst-case examples (8); see Algorithm 1 for the full description of our algorithm.

**Ensembles for classification** The hyperparameter $\gamma$—which is inversely proportional to $\rho$, the distance between the fictitious target distribution and the source—controls the ability to generalize outside the source domain. Since target domains are unknown, it is difficult to choose an appropriate level of $\gamma$ a priori. We propose a heuristic ensemble approach where we train $s$ models $\left\{\theta^0, ..., \theta^s\right\}$. Each model is associated with a different value of $\gamma$, and thus to fictitious target distributions with varying distances from the source $P_0$. To select the best model at test time—inspired by Hendrycks and Gimpel [13]—given a sample $x$, we select the model $\theta^{u^\star(x)}$ with the greatest softmax score

$$u^\star(x) := \arg\max_{1 \leq u \leq s} \max_{1 \leq j \leq k} \theta_{c,j}^{u\top} g(\theta_f^u; x). \tag{9}$$

## 3 Theoretical Motivation

In our iterative algorithm (Algorithm 1), the *maximization* phase (8) was a key step that augmented the dataset with adversarially perturbed data points, which was followed by standard stochastic gradient updates to the model parameters. In this section, we provide some theoretical understanding of the augmentation step (8). First, we show that the augmented data points (8) can be interpreted as *Tikhonov regularized Newton-steps* [21, 25] in the semantic space under the current model. Roughly speaking, this gives the sense in which Algorithm 1 is an adaptive data augmentation algorithm that adds data points from fictitious "hard" target distributions. Secondly, recall the robust surrogate loss (5) whose stochastic gradients were used to update the model parameters $\theta$ in the *minimization* step (Eq (7)). In the classification setting, we show that the robust surrogate (5) roughly corresponds to a novel data-dependent regularization scheme on the softmax loss $\ell$. Instead of penalizing towards zero like classical regularizers (*e.g.*, ridge or lasso), our data-dependent regularization term penalizes deviations from the parameter vector corresponding to that of the true label.

### 3.1 Adaptive Data Augmentation

We now give an interpretation for the augmented data points in the maximization phase (8). Concretely, we fix $\theta \in \Theta$, $x_0 \in \mathcal{X}$, $y_0 \in \mathcal{Y}$, and consider an $\epsilon$-maximizer

$$x_\epsilon^\star \in \epsilon\text{-}\arg\max_{x \in \mathcal{X}} \left\{\ell(\theta; (x, y_0)) - \gamma c_\theta((x, y_0), (x_0, y_0))\right\}.$$

We let $z_0 := g(\theta_f; x_0) \in \mathbb{R}^p$, and abuse notation by using $\ell(\theta; (z_0, y_0)) := \ell(\theta; (x_0, y_0))$. In what follows, we show that the feature mapping $g(\theta_f; x_\epsilon^\star)$ satisfies

$$g(\theta_f; x_\epsilon^\star) = \underbrace{g(\theta_f; x_0) + \frac{1}{\gamma} \left(I - \frac{1}{\gamma}\nabla_{zz}\ell(\theta; (z_0, y_0))\right)^{-1} \nabla_z \ell(\theta; (z_0, y_0))}_{=: \, \widehat{g}_{\text{newton}}(\theta_f; x_0)} + O\left(\sqrt{\frac{\epsilon}{\gamma}} + \frac{1}{\gamma^2}\right).$$
$$\tag{10}$$

Intuitively, this implies that the adversarially perturbed sample $x_\epsilon^\star$ is drawn from a fictitious target distribution where probability mass on $z_0 = g(\theta_f; x_0)$ was transported to $\widehat{g}_{\text{newton}}(\theta_f; x_0)$. We note that the transported point in the semantic space corresponds to a *Tikhonov regularized Newton-step* [21, 25] on the loss $z \mapsto \ell(\theta; (z, y_0))$ at the current model $\theta$. Noting that computing $\widehat{g}_{\text{newton}}(\theta_f; x_0)$ involves backsolves on a large dense matrix, we can interpret our gradient ascent updates in the maximization phase (8) as an iterative scheme for approximating this quantity.

We assume sufficient smoothness, where we use $\|H\|$ to denote the $\ell_2$-operator norm of a matrix $H$.

**Assumption 1.** *There exists $L_0, L_1 > 0$ such that, for all $z, z' \in \mathbb{R}^p$, we have $|\ell(\theta; (z, y_0)) - \ell(\theta; (z', y_0))| \leq L_0 \|z - z'\|_2$ and $\|\nabla_z \ell(\theta; (z, y_0)) - \nabla_z \ell(\theta; (z', y_0))\|_2 \leq L_1 \|z - z'\|_2$.*

**Assumption 2.** *There exists $L_2 > 0$ such that, for all $z, z' \in \mathbb{R}^p$, we have $\|\nabla_{zz}\ell(\theta; (z, y_0)) - \nabla_{zz}\ell(\theta; (z', y_0))\| \leq L_2 \|z - z'\|_2$.*

Then, we have the following bound (10) whose proof we defer to Appendix A.1.

**Theorem 1.** *Let Assumptions 1, 2 hold. If $Im(g(\theta_f; \cdot)) = \mathbb{R}^p$ and $\gamma > L_1$, then*

$$\|g(\theta_f; x_\epsilon^\star) - \widehat{g}_{\text{newton}}(\theta_f; x_0)\|_2^2 \leq \frac{2\epsilon}{\gamma - L_1} + \frac{L_2}{3(\gamma - L_1)}\left\{\left(\frac{5L_0}{\gamma}\right)^3 + \left(\frac{L_0}{\gamma - L_1}\right)^3 + \left(\frac{2\epsilon}{\gamma}\right)^{\frac{3}{2}}\right\}.$$

## 3.2 Data-Dependent Regularization

In this section, we argue that under suitable conditions on the loss,

$$\phi_\gamma(\theta; (z, y)) = \ell(\theta; (z, y)) + \frac{1}{\gamma} \|\nabla_z \ell(\theta; (z, y))\|_2^2 + O\left(\frac{1}{\gamma^2}\right).$$

For classification problems, we show that the robust surrogate loss (5) corresponds to a particular data-dependent regularization scheme. Let $\ell(\theta; (x, y))$ be the $m$-class softmax loss (2) given by

$$\ell(\theta; (x, y)) = -\log p_y(\theta, x) \ \text{ where } \ p_j(\theta, x) := \frac{\exp(\theta_{c,j}^\top g(\theta, x))}{\sum_{l=1}^m \exp(\theta_{c,l}^\top g(\theta_f; x))}.$$

where $\theta_{c,j} \in \mathbb{R}^p$ is the $j$-th row of the classification layer weight $\theta_c \in \mathbb{R}^{p \times m}$. Then, the robust surrogate $\phi_\gamma$ is an approximate regularizer on the classification layer weights $\theta_c$

$$\phi_\gamma(\theta; (x, y)) = \ell(\theta; (x, y)) + \frac{1}{\gamma} \left\| \theta_{c,y} - \sum_{j=1}^m p_j(\theta, x)\theta_{c,j} \right\|_2^2 + O\left(\frac{1}{\gamma^2}\right). \tag{11}$$

The expansion (11) shows that the robust surrogate (5) is roughly equivalent to data-dependent regularization where we minimize the distance between $\sum_{j=1}^m p_j(\theta, x)\theta_{c,j}$, our "average estimated linear classifier", to $\theta_{c,y}$, the linear classifier corresponding to the true label $y$. Concretely, for any fixed $\theta \in \Theta$, we have the following result where we use $L(\theta) := 2\max_{1 \le j' \le m} \|\theta_{c,j'}\|_2 \sum_{j=1}^m \|\theta_{c,j}\|_2$ to ease notation. See Appendix A.3 for the proof.

**Theorem 2.** *If $Im(g(\theta_f; \cdot)) = \mathbb{R}^p$ and $\gamma > L(\theta)$, the softmax loss (2) satisfies*

$$\frac{1}{\gamma + L(\theta)} \left\| \theta_{c,y} - \sum_{j=1}^m p_j(\theta, x)\theta_{c,j} \right\|_2^2 \le \phi_\gamma(\theta, (x, y)) - \ell(\theta, (x, y)) \le \frac{1}{\gamma - L(\theta)} \left\| \theta_{c,y} - \sum_{j=1}^m p_j(\theta, x)\theta_{c,j} \right\|_2^2.$$

## 4 Experiments

We evaluate our method for both classification and semantic segmentation settings, following the evaluation scenarios of domain adaptation techniques [9, 39, 14], though in our case the target domains are unknown at training time. We summarize our experimental setup including implementation details, evaluation metrics and datasets for each task.

**Digit classification**   We train on MNIST [19] dataset and test on MNIST-M [9], SVHN [30], SYN [9] and USPS [6]. We use $10,000$ digit samples for training and evaluate our models on the respective test sets of the different target domains, using accuracy as a metric. In order to work with comparable datasets, we resized all the images to $32 \times 32$, and treated images from MNIST and USPS as RGB. We use a ConvNet [18] with architecture *conv-pool-conv-pool-fc-fc-softmax* and set the hyperparameters $\alpha = 0.0001$, $\eta = 1.0$, $T_{\min} = 100$ and $T_{\max} = 15$. In the minimization phase, we use Adam [17] with batch size equal to $32^4$. We compare our method against the Empirical Risk Minimization (ERM) baseline and different regularization techniques (Dropout [35], ridge).

**Semantic scene segmentation**   We use the SYTHIA[31] dataset for semantic segmentation. The dataset contains images from different locations (we use *Highway*, *New York-like City* and *Old European Town*), and different weather/time/date conditions (we use *Dawn*, *Fog*, *Night*, *Spring* and *Winter*. We train models on a source domain and test on other domains, using the standard mean Intersection Over Union *(mIoU)* metric to evaluate our performance [8]. We arbitrarily chose images from the left front camera throughout our experiments. For each one, we sample 900 random images (resized to $192 \times 320$ pixels) from the training set. We use a Fully Convolutional Network (FCN) [23], with a ResNet-50 [11] body and set the hyperparameters $\alpha = 0.0001$, $\eta = 2.0$, $T_{\min} = 500$ and $T_{\max} = 50$. For the minimization phase, we use Adam [17] with batch size equal to $8$. We compare our method against the ERM baseline.

### 4.1 Results on Digit Classification

In this section, we present and discuss the results on the digit classification experiment. Firstly, we are interested in analyzing the role of the semantic constraint we impose. Figure 1a *(top)* shows

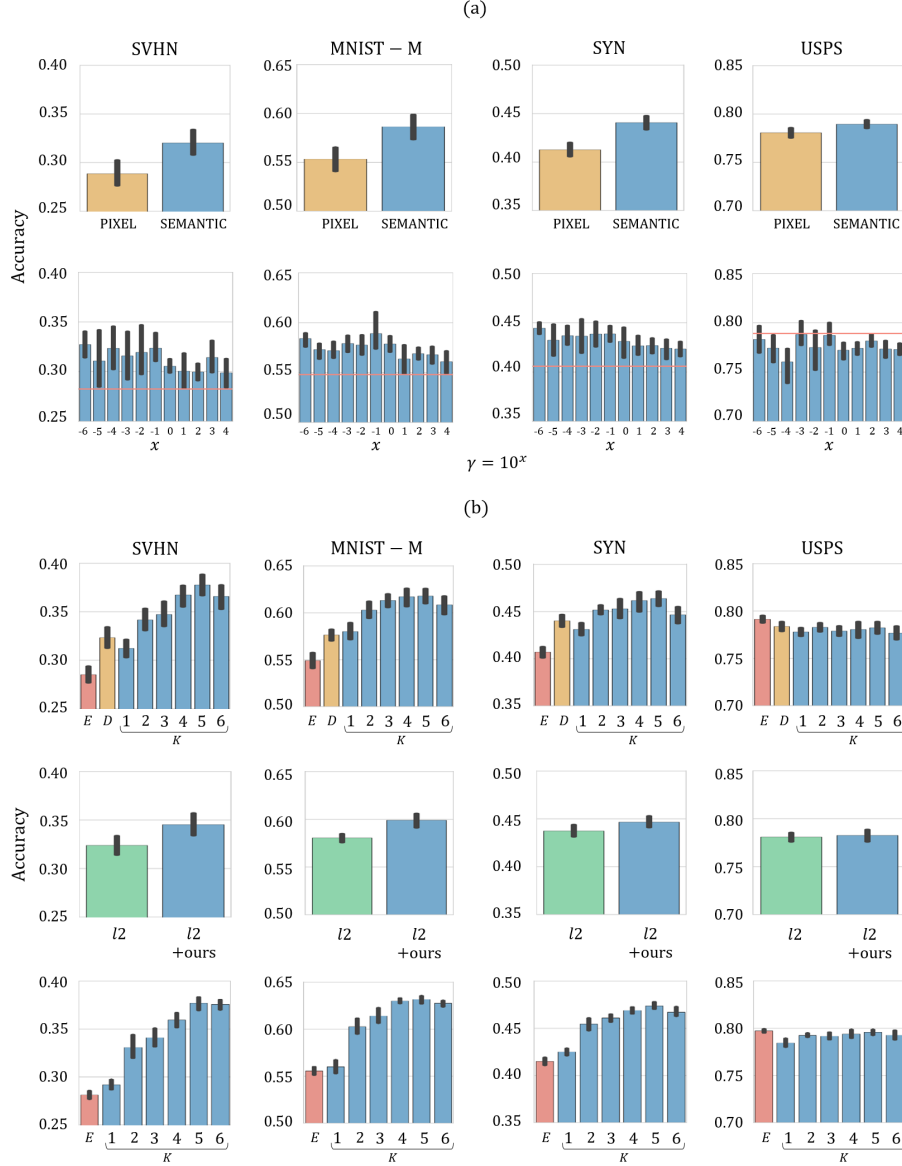

**Figure 1.** Results associated with models trained with $10,000$ MNIST samples and tested on SVHN, MNIST-M, SYN and USPS ($1^{st}$, $2^{nd}$, $3^{rd}$ and $4^{th}$ columns, respectively). *Panel (a), top:* comparison between distances in the pixel space (yellow) and in the semantic space (blue), with $\gamma = 10^4$ and $K = 1$. *Panel (a), bottom:* comparison between our method with $K = 2$ and different $\gamma$ values (blue bars) and ERM (red line). *Panel (b), top:* comparison between our method with $\gamma = 1.0$ and different number of iterations $K$ (blue), ERM (red) and Dropout [35] (yellow). *Panel (b), middle:* comparison between models regularized with ridge (green) and with ridge + our method with $\gamma = 1.0$ and $K = 1$ (blue). *Panel (b), bottom:* results related to the ensemble method, using models trained with our methods with different number of iterations $K$ (blue) and using models trained via ERM (red). The reported results are obtained by averaging over 10 different runs; black bars indicate the range of accuracy spanned.

performances associated with models trained with Algorithm 1 with $K = 1$ and $\gamma = 10^4$, with the constraint in the semantic space (as discussed in Section 2) and in the pixel space [34] (blue and yellow bars, respectively). Figure 1a *(bottom)* shows performances of models trained with our method using different values of the hyperparameter $\gamma$ (with $K = 2$) and with ERM (blue bars and red lines, respectively). These plots show (i) that moving the constraint on the semantic space carries benefits when models are tested on unseen domains and (ii) that models trained with Algorithm 1 outperform models train with ERM for any value of $\gamma$ on out-of-sample domains (SVHN, MNIST-M

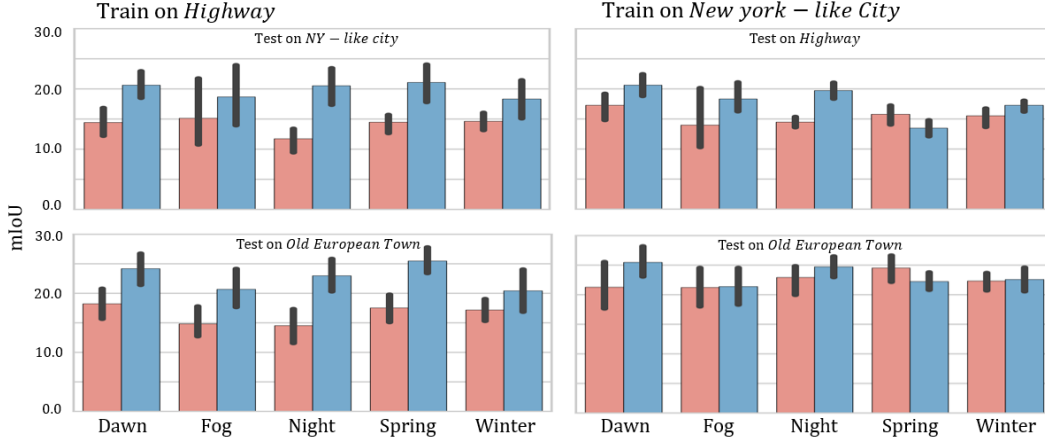

**Figure 2.** Results obtained with semantic segmentation models trained with ERM *(red)* and our method with $K = 1$ and $\gamma = 1.0$ *(blue)*. *Leftmost* panels are associated with models trained on *Highway*, *rightmost* panels are associated with models trained on *New York-like City*. Test datasets are *Highway*, *New York-like City* and *Old European Town*.

and SYN). The latter result is a rather desired achievement, since this hyperparameter cannot be properly cross-validated. On USPS, our method causes accuracy to drop since MNIST and USPS are very similar datasets, thus the image domain that USPS belongs to is not explored by our algorithm during the training procedure, which optimizes for worst case performance.

Figure 1b *(top)* reports results related to models trained with our method (blue bars), varying the number of iterations $K$ and fixing $\gamma = 1.0$, and results related to ERM (red bars) and Dropout [35] (yellow bars). We observe that our method improves performances on SVHN, MNIST-M and SYN, outperforming both ERM and Dropout [35] statistically significantly. In Figure 1b *(middle)*, we compare models trained with ridge regularization (green bars) with models trained with Algorithm 1 (with $K = 1$ and $\gamma = 1.0$) and ridge regularization (blue bars); these results show that our method can potentially benefit from other regularization approaches, as in this case we observed that the two effects sum up. We further report in Appendix B a comparison between our method and an unsupervised domain adaptation algorithm (ADDA [39]), and results associated with different values of the hyperparameters $\gamma$ and $K$.

Finally, we report the results obtained by learning an ensemble of models. Since the hyperparameter $\gamma$ is nontrivial to set a priori, we use the softmax confidences (9) to choose which model to use at test time. We learn ensemble of models, each of which is trained by running Algorithm 1 with different values of the $\gamma$ as $\gamma = 10^{-i}$, with $i = \{0, 1, 2, 3, 4, 5, 6\}$. Figure 1b *(bottom)* shows the comparison between our method with different numbers of iterations $K$ and ERM (blue and red bars, respectively). In order to separate the role of ensemble learning, we learn an ensemble of baseline models each corresponding to a different initialization. We fix the number of models in the ensemble to be the same for both the baseline (ERM) and our method. Comparing Figure 1b *(bottom)* with Figure 1b *(top)* and Figure 1a *(bottom)*, our ensemble approach achieves higher accuracy in different testing scenarios. We observe that our out-of-sample performance improves as the number of iterations $K$ gets large. Also in the ensemble setting, for the USPS dataset we do not see any improvement, which we conjecture to be an artifact of the trade-off between good performance on domains far away from training, and those closer.

## 4.2   Results on Semantic Scene Segmentation

We report a comparison between models trained with ERM and models trained with our method (Algorithm 1 with $K = 1$). We set $\gamma = 1.0$ in every experiment, but stress that this is an arbitrary value; we did not observe a strong correlation between the different values of $\gamma$ and the general behavior of the models in this case. Its role was more meaningful in the ensemble setting where each model is associated with a different level of robustness, as discussed in Section 2. In this setting, we do not apply the ensemble approach, but only evaluate the performances of the single models. The

main reason for this choice is the fact that the heuristics developed to choose the correct model at test time in effect cannot be applied in a straightforward fashion to a semantic segmentation problem.

Figure 2 reports numerical results obtained. Specifically, leftmost plots report results associated with models trained on sequences from the *Highway* split and tested on the *New York-like City* and the *Old European Town* splits (*top-left* and *bottom-left*, respectively); rightmost plots report results associated with models trained on sequences from the *New York-like City* split and tested on the *Highway* and the *Old European Town* splits (*top-right* and *bottom-right*, respectively). The training sequences (*Dawn*, *Fog*, *Night*, *Spring* and *Winter*) are indicated on the x-axis. Red and blue bars indicate average mIoUs achieved by models trained with ERM and by models trained with our method, respectively. These results were calculated by averaging over the mIoUs obtained with each model on the different conditions of the test set. As can be observed, models trained with our method mostly better generalize to unknown data distributions. In particular, our method always outperforms the baseline by a statistically significant margin when the training images are from *Night* scenarios. This is since the baseline models trained on images from *Night* are strongly biased towards dark scenery, while, as a consequence of training over worst-case distributions, our models can overcome this strong bias and better generalize across different unseen domains.

## 5   Conclusions and Future Work

We study a new adversarial data augmentation procedure that learns to better generalize across unseen data distributions, and define an ensemble method to exploit this technique in a classification framework. This is in contrast to domain adaptation algorithms, which require a sufficient number of samples from a known, a priori fixed target distribution. Our experimental results show that our iterative procedure provides broad generalization behavior on digit recognition and cross-season and cross-weather semantic segmentation tasks.

For future work, we hope to extend the ensemble methods by defining novel decision rules. The proposed heuristics (9) only apply to classification settings, and extending them to a broad realm of tasks including semantic segmentation is an important direction. Many theoretical questions still remain. For instance, quantifying the behavior of data-dependent regularization schemes presented in Section 3 would help us better understand adversarial training methods in general.

## Footnotes

[3]By *semantic space* we mean learned representations since recent works [7, 16] suggest that distances in the space of learned representations of high capacity models typically correspond to semantic distances in visual space.

[4]Models were implemented using Tensorflow, and training procedures were performed on NVIDIA GPUs. Code is available at `https://github.com/ricvolpi/generalize-unseen-domains`

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
