[Supplementary Material]

# A Proofs

## A.1 Proof of Theorem 1

Recall that we consider a fixed $\theta \in \Theta$, $x_0 \in \mathcal{X}$, $y_0 \in \mathcal{Y}$, and $z_0 = g(\theta_f; x_0)$. We begin by noting that since $\mathrm{Im}(g(\theta_f; \cdot)) = \mathbb{R}^p$, we have

$$\phi_\gamma(\theta; (x_0, y_0)) = \sup_{x \in \mathcal{X}} \left\{ \ell(\theta; (x, y_0)) - \gamma c_\theta((x, y_0), (x_0, y_0)) \right\}$$

$$= \sup_{z \in \mathbb{R}^p} \left\{ \ell(\theta; (z, y_0)) - \frac{\gamma}{2} \|z - z_0\|_2^2 =: h(z) \right\}. \tag{12}$$

Similarly as $x_\epsilon^\star$, let $z_\epsilon^\star$ be an $\epsilon$-optimizer to the problem (12)

$$z_\epsilon^\star \in \epsilon\text{-}\arg\max_{z \in \mathbb{R}^p} \left\{ \ell(\theta; (z, y_0)) - \frac{\gamma}{2} \|z - z_0\|_2^2 \right\}.$$

To further ease notation, let us denote

$$\ell_1(\theta; (z, y_0)) := \ell(\theta; (z_0, y_0)) + \nabla_z \ell(\theta; (z_0, y_0))^\top (z - z_0)$$

$$\ell_2(\theta; (z, y_0)) := \ell(\theta; (z_0, y_0)) + \nabla_z \ell(\theta; (z_0, y_0))^\top (z - z_0) + \frac{1}{2}(z - z_0)^\top \nabla_{zz} \ell(\theta; (z_0, y_0))(z - z_0),$$

the first- and second-order approximation of $z \mapsto \ell(\theta; (z, y_0))$ around $z = z_0$ respectively.

First, we note that $\|\nabla_{zz} \ell(\theta; (z, y))\| \leq L_1 < \gamma$ by hypothesis and hence, $\widehat{g}_{\mathrm{newton}}(\theta_f; x_0)$ attains the maximum in the problem

$$\widehat{g}_{\mathrm{newton}}(\theta_f; x_0) = z_0 + \frac{1}{\gamma} \left( I - \frac{1}{\gamma} \nabla_{zz} \ell(\theta; (z_0, y_0)) \right)^{-1} \nabla_z \ell(\theta; (z_0, y_0)) \tag{13}$$

$$= \arg\max_{z \in \mathbb{R}^p} \left\{ \ell_2(\theta; (z, y_0)) - \frac{\gamma}{2} \|z - z_0\|_2^2 := h_2(z) \right\}$$

Now, note that $h_2(z) = \ell_2(\theta; (z, y_0)) - \frac{\gamma}{2} \|z - z_0\|_2^2$ is $(\gamma - L_1)$ - strongly concave since

$$\lambda_{\min}(-\nabla_{zz} h(z)) \geq \gamma - \lambda_{\max}(\nabla_{zz} \ell_2(\theta; (z, y_0))) \geq \gamma - L_1$$

by Assumption 1, where $\lambda_{\max}$ and $\lambda_{\min}$ denotes the maximum and minimum eigenvalue respectively. Recalling the definition of $h(z)$ given in Eq (12), we then have

$$\frac{\gamma - L_1}{2} \|z_\epsilon^\star - \widehat{g}_{\mathrm{newton}}(\theta_f; x_0)\|_2^2 \leq h_2(z_\epsilon^\star) - h_2(\widehat{g}_{\mathrm{newton}}(\theta_f; x_0))$$

$$= h(z_\epsilon^\star) - h(\widehat{g}_{\mathrm{newton}}(\theta_f; x_0)) + h_2(z_\epsilon^\star) - h(z_\epsilon^\star)$$

$$+ h(\widehat{g}_{\mathrm{newton}}(\theta_f; x_0)) - h_2(\widehat{g}_{\mathrm{newton}}(\theta_f; x_0))$$

$$\leq \epsilon + h_2(z_\epsilon^\star) - h(z_\epsilon^\star)$$

$$+ h(\widehat{g}_{\mathrm{newton}}(\theta_f; x_0)) - h_2(\widehat{g}_{\mathrm{newton}}(\theta_f; x_0)) \tag{14}$$

where we used the definition of $z_\epsilon^\star$ in the last inequality.

Next, we note that $h_2$ and $h$ are close by Taylor expansion.

**Lemma 2** ([29, Lemma 1]). *Let $f : \mathbb{R}^p \to \mathbb{R}$ have a $L$-Lipschitz Hessian so that for all $z, z' \in \mathbb{R}^p$, $\|\nabla_{zz} f(z) - \nabla_{zz} f(z')\| \leq L \|z - z'\|_2$. Then, for all $z, z' \in \mathbb{R}^p$,*

$$\left| f(z') - f(z) - \nabla f(z)^\top (z' - z) - \frac{1}{2}(z' - z)^\top \nabla_{zz} f(z)(z' - z) \right| \leq \frac{L}{6} \|z' - z\|_2^2.$$

Applying Lemma 2, we have that

$$|h_2(z) - h(z)| \leq \frac{L_2}{6} \|z - z_0\|_2^3.$$

Using this inequality in the bound (14), we arrive at

$$\frac{\gamma - L_1}{2} \|z_\epsilon^\star - \widehat{g}_{\mathrm{newton}}(\theta_f; x_0)\|_2^2$$

$$\leq \epsilon + \frac{L_2}{6} \left( \|z_0 - z_\epsilon^\star\|_2^3 + \|z_0 - \widehat{g}_{\mathrm{newton}}(\theta_f; x_0)\|_2^3 \right) \tag{15}$$

From definition (13) of $\widehat{g}_{\text{newton}}(\theta_f; x_0)$, we have

$$\|z_0 - \widehat{g}_{\text{newton}}(\theta_f; x_0)\|_2^3 \leq \left(\frac{1}{\gamma}\right)^3 \left(\frac{\gamma}{\gamma - L_1}\right)^3 L_0^3. \qquad (16)$$

Next, to bound $\|z_0 - z_\epsilon^\star\|_2$ in the bound (15), we show that $z_\epsilon^\star$ and $z_0$ are at most $O(1/\gamma)$-away. We defer the proof of the following lemma to Appendix A.2

**Lemma 3.** *Let Assumption 1 hold and $Im(g(\theta_f; \cdot)) = \mathbb{R}^p$. Then,*

$$\left\|z_\epsilon^\star - z_0 - \frac{1}{\gamma}\nabla_z \ell(\theta; (z_0, y_0))\right\|_2 \leq \frac{4L_0}{\gamma} + \sqrt{\frac{2\epsilon}{\gamma}}.$$

Applying Lemma 3 to bound $\|z_0 - z_\epsilon^\star\|_2^3$ on the right hand side of inequality (15), and using the bound (16) for $\|z_0 - \widehat{g}_{\text{newton}}(\theta_f; x_0)\|_2^3$, we obtain

$$\frac{\gamma - L_1}{2}\|z_\epsilon^\star - \widehat{g}_{\text{newton}}(\theta_f; x_0)\|_2^2 \leq \epsilon + \frac{L_2}{6}\left[\left(\frac{5L_0}{\gamma}\right)^3 + \left(\frac{2\epsilon}{\gamma}\right)^{\frac{3}{2}} + \left(\frac{L_0}{\gamma - L_1}\right)^3\right].$$

This gives the final result.

## A.2 Proof of Lemma 3

We use the following key lemma which says that for functions that satisfy a growth condition, its minimum is stable to perturbations to the function.

**Lemma 4** ([4, Proposition 4.32]). *Suppose that $f_0$ satisfies the second-order growth condition: there exists a $c > 0$ such that if we denote by $z^\star$ the minimizer of $f$ so that $f_0(z^\star) = \inf_{z \in \mathbb{R}^p} f_0(z)$, we have for all $z$*

$$f_0(z) \geq f_0(z^\star) + c\|z - z^\star\|_2^2.$$

*If there is a function $f_1 : \mathbb{R}^p \to \mathbb{R}$ such that $f_0 - f_1$ is $\kappa$-Lipschitz on a neighborhood $N$ of $x^\star$, then $z$, any $\epsilon$-approximate minimizer of $f_1$ in $N$, satisfies*

$$\|z - z^\star\|_2 \leq c^{-1}\kappa + c^{-1/2}\epsilon^{1/2}$$

Letting $f_0(z) := -\ell_1(\theta; (z, y_0)) + \frac{\gamma}{2}\|z - z_0\|_2^2$ and $f_1(z) := -h(z) = -\ell(\theta; (z, y_0)) + \frac{\gamma}{2}\|z - z_0\|_2^2$, note first that $f_0$ is $\gamma$-strongly convex. Further, $f_0(z) - f_1(z) = \ell(\theta; (z, y_0)) - \ell_1(\theta; (z, y_0))$ is $2L_0$-Lipschitz by Assumption 1. Applying Lemma 4, we obtain the result.

## A.3 Proof of Theorem 2

Again, we abuse notation by writing $\ell(\theta; (z, y)) = \ell(\theta; (x, y))$ for $z = g(\theta_f; x) \in \mathbb{R}^p$, and similarly $p_j(\theta; z)$ and $\phi_\gamma(\theta; z)$. We begin by noting that since $Im(g(\theta, \cdot)) = \mathbb{R}^p$, we have

$$\phi_\gamma(\theta; (x, y)) = \sup_{z' \in \mathbb{R}^p} \left\{\ell(\theta; (z', y)) - \frac{\gamma}{2}\|z - z'\|_2^2\right\}.$$

The following claim will be crucial.

**Claim 5.** *If $z \mapsto \nabla_z \ell(\theta; (z, y))$ is $L$-Lipschitz with respect to the $\|\cdot\|_2$-norm, then*

$$\frac{1}{\gamma + L}\|\nabla_z \ell(\theta; (z, y))\|_2^2 \leq \phi_\gamma(\theta; (z, y)) - \ell(\theta; (z, y)) \leq \frac{1}{\gamma - L}\|\nabla_z \ell(\theta; (z, y))\|_2^2.$$

**Proof of Claim** From Taylor's theorem, we have

$$\left|\ell(\theta; (z', y)) - \ell(\theta; (z, y)) - \nabla_z \ell(\theta; (z, y))^\top (z' - z)\right| \leq \frac{1}{2}L\|z - z'\|_2^2.$$

Using this approximation in the definition of $\phi_\gamma(\theta; (z, y))$, we get

$$\phi_\gamma(\theta; (z, y)) \leq \sup_{z'}\left\{\ell(\theta; (z, y)) + \nabla_z \ell(\theta; (z, y))^\top (z' - z) - \frac{\gamma - L}{2}\|z - z'\|_2^2\right\}$$

$$= \ell(\theta; (z, y)) + \frac{1}{2(\gamma - L)}\|\nabla_z \ell(\theta; (z, y))\|_2^2.$$

Similarly, we can compute the lower bound

$$\phi_\gamma(\theta; (z, y)) \geq \sup_{z'} \left\{ \ell(\theta; (z, y)) + \nabla_z \ell(\theta; (z, y))^\top (z - z') - \frac{\gamma + L}{2} \|z - z'\|_2^2 \right\}$$

$$= \ell(\theta; (z, y)) + \frac{1}{2(\gamma + L)} \|\nabla_z \ell(\theta; (z, y))\|_2^2 .$$

Combining the two bounds, the claim follows. □

From the claim, it suffices to show that $z \mapsto \nabla_z \ell(\theta; (z, y))$ is $L$-Lipschitz. From $\nabla_z \ell(\theta; (z, y)) = -\theta_{c,y} + \sum_{j=1}^m p_j(\theta; z)\theta_{c,j}$, we have

$$\|\nabla_z \ell(\theta; (z', y)) - \nabla_z \ell(\theta; (z, y))\|_2 = \left\| \sum_{j=1}^m (p_j(\theta; z) - p_j(\theta; z'))\theta_j \right\|_2 .$$

Now, since

$$\|\nabla_z p_j(\theta; z)\|_2 = \left\| -p_j(\theta; z) \left( \theta_j - \sum_{l=1}^m p_l(\theta; z)\theta_l \right) \right\|_2 \leq 2 \max_{1 \leq j \leq m} \|\theta_{c,j}\|_2 ,$$

we conclude that

$$\|\nabla_z \ell(\theta; (z', y)) - \nabla_z \ell(\theta; (z, y))\|_2 \leq L(\theta) \|z - z'\|_2 .$$

**Figure 3.** Results obtained by running ADDA algorithm [39] using $10,000$ labeled MNIST samples and a number of target samples indicated on the x-axis. The *blue* lines indicate results obtained with our method with $K = 2$ and $\gamma = 1.0$. Test sets are MNIST-M *(left)*, SYN *(middle)* and USPS *(right)*.

# B  Additional Experimental Results

Table 1 reports results associated with the digit experiment (Section 4.1, Figure 2). In particular, it reports numerical results (averaged over $10$ different runs) obtained with models trained with Algorithm 1 by varying the hyperparameters $K$ and $\gamma$. Training set is constituted by $10,000$ MNIST samples, models were tested on SVHN, MNIST-M, SYN and USPS (see Figure 1 *(top)*). The baselines (accuracies achieved by models trained with ERM) are:

- SVHN: $0.283 \pm 0.032$
- MNIST-M: $0.548 \pm 0.021$
- SYN: $0.406 \pm 0.022$
- USPS: $0.789 \pm 0.017$

Table 2 reports results associated with the semantic segmentation experiment (Section 4.2, Figure 3). To summarize, it reports results obtained by training models on *Highway* and testing them on *New York-like City* and *Old European Town*, and by training models on *New York-like City* and testing them on *Highway* and *Old European Town* (see Figure 1 *(bottom)* to observe the different weather/time/date conditions). The comparison is between models trained with ERM *(ERM rows)* and our method *(Ours rows)*, *e.g.* Algorithm 1 with $K = 1$ and $\gamma = 1.0$.

Finally, Figure 4 reports a comparison between our method *(blue)* and the unsupervised domain adaptation algorithm ADDA [39] *(yellow)*, by varying the number of target images fed to the latter during training. Note that, since unsupervised domain adaptation algorithms make use of target data during training while our method does not, the comparison is not fair. However, we are interested in evaluating to which extent our method can compete with a well performing unsupervised domain adaptation algorithm [39]. While on MNIST $\rightarrow$ USPS split ADDA clearly outperforms our method, on MNIST $\rightarrow$ MNIST-M the accuracies reached by our method are just slightly lower than the ones reached by ADDA, and on MNIST $\rightarrow$ SYN our method outperforms it, even if the domain adaptation algorithm has access to a large number of samples from the target domain. Finally, note that MNIST $\rightarrow$ SVHN results are not provided because ADDA would not converge on this split (in effect, these results are neither reported in the original work [39]). Instead, models trained on MNIST samples using our method better generalize to SVHN, as shown in Section 4.1.

**Table 1.** Results obtained by training models with Algorithm 1 on 10, 000 MNIST samples and testing them on SVHN, MNIST-M, SYN and USPS. Results are averaged over 20 different runs.

| | K=1 | K=2 | K=3 | K=4 |
|---|---|---|---|---|
| **SVHN** | | | | |
| $\gamma = 10^{-6}$ | $0.287 \pm 0.006$ | $0.327 \pm 0.016$ | $0.334 \pm 0.031$ | $0.328 \pm 0.033$ |
| $\gamma = 10^{-5}$ | $0.284 \pm 0.036$ | $0.311 \pm 0.033$ | $0.316 \pm 0.036$ | $0.331 \pm 0.026$ |
| $\gamma = 10^{-4}$ | $0.331 \pm 0.018$ | $0.324 \pm 0.026$ | $0.336 \pm 0.020$ | $0.325 \pm 0.030$ |
| $\gamma = 10^{-3}$ | $0.294 \pm 0.023$ | $0.316 \pm 0.029$ | $0.309 \pm 0.024$ | $0.343 \pm 0.017$ |
| $\gamma = 10^{-2}$ | $0.290 \pm 0.041$ | $0.320 \pm 0.030$ | $0.341 \pm 0.030$ | $0.346 \pm 0.033$ |
| $\gamma = 10^{-1}$ | $0.284 \pm 0.007$ | $0.324 \pm 0.017$ | $0.307 \pm 0.026$ | $0.323 \pm 0.029$ |
| $\gamma = 10^{0}$ | $0.284 \pm 0.012$ | $0.306 \pm 0.008$ | $0.314 \pm 0.022$ | $0.335 \pm 0.029$ |
| $\gamma = 10^{1}$ | $0.305 \pm 0.031$ | $0.301 \pm 0.035$ | $0.316 \pm 0.027$ | $0.343 \pm 0.030$ |
| $\gamma = 10^{2}$ | $0.304 \pm 0.032$ | $0.300 \pm 0.017$ | $0.327 \pm 0.026$ | $0.321 \pm 0.034$ |
| $\gamma = 10^{3}$ | $0.289 \pm 0.030$ | $0.314 \pm 0.032$ | $0.300 \pm 0.017$ | $0.304 \pm 0.025$ |
| $\gamma = 10^{4}$ | $0.300 \pm 0.020$ | $0.299 \pm 0.028$ | $0.325 \pm 0.015$ | $0.340 \pm 0.026$ |
| **MNIST-M** | | | | |
| $\gamma = 10^{-6}$ | $0.561 \pm 0.013$ | $0.584 \pm 0.008$ | $0.581 \pm 0.009$ | $0.588 \pm 0.013$ |
| $\gamma = 10^{-5}$ | $0.564 \pm 0.024$ | $0.573 \pm 0.010$ | $0.573 \pm 0.024$ | $0.589 \pm 0.017$ |
| $\gamma = 10^{-4}$ | $0.583 \pm 0.011$ | $0.572 \pm 0.010$ | $0.586 \pm 0.015$ | $0.578 \pm 0.031$ |
| $\gamma = 10^{-3}$ | $0.562 \pm 0.026$ | $0.579 \pm 0.010$ | $0.567 \pm 0.023$ | $0.601 \pm 0.018$ |
| $\gamma = 10^{-2}$ | $0.539 \pm 0.037$ | $0.578 \pm 0.013$ | $0.590 \pm 0.014$ | $0.598 \pm 0.014$ |
| $\gamma = 10^{-1}$ | $0.556 \pm 0.017$ | $0.589 \pm 0.021$ | $0.576 \pm 0.018$ | $0.576 \pm 0.019$ |
| $\gamma = 10^{0}$ | $0.557 \pm 0.017$ | $0.579 \pm 0.009$ | $0.571 \pm 0.010$ | $0.584 \pm 0.024$ |
| $\gamma = 10^{1}$ | $0.568 \pm 0.022$ | $0.564 \pm 0.028$ | $0.579 \pm 0.024$ | $0.589 \pm 0.016$ |
| $\gamma = 10^{2}$ | $0.564 \pm 0.025$ | $0.569 \pm 0.013$ | $0.579 \pm 0.019$ | $0.578 \pm 0.021$ |
| $\gamma = 10^{3}$ | $0.558 \pm 0.016$ | $0.568 \pm 0.017$ | $0.568 \pm 0.010$ | $0.567 \pm 0.021$ |
| $\gamma = 10^{4}$ | $0.567 \pm 0.022$ | $0.561 \pm 0.023$ | $0.570 \pm 0.015$ | $0.579 \pm 0.016$ |
| **SYN** | | | | |
| $\gamma = 10^{-6}$ | $0.415 \pm 0.013$ | $0.445 \pm 0.007$ | $0.440 \pm 0.012$ | $0.443 \pm 0.013$ |
| $\gamma = 10^{-5}$ | $0.409 \pm 0.029$ | $0.432 \pm 0.020$ | $0.437 \pm 0.024$ | $0.443 \pm 0.014$ |
| $\gamma = 10^{-4}$ | $0.439 \pm 0.011$ | $0.437 \pm 0.011$ | $0.446 \pm 0.018$ | $0.440 \pm 0.022$ |
| $\gamma = 10^{-3}$ | $0.417 \pm 0.018$ | $0.437 \pm 0.021$ | $0.436 \pm 0.017$ | $0.450 \pm 0.010$ |
| $\gamma = 10^{-2}$ | $0.417 \pm 0.022$ | $0.439 \pm 0.015$ | $0.447 \pm 0.020$ | $0.450 \pm 0.014$ |
| $\gamma = 10^{-1}$ | $0.405 \pm 0.011$ | $0.439 \pm 0.009$ | $0.438 \pm 0.018$ | $0.439 \pm 0.021$ |
| $\gamma = 10^{0}$ | $0.418 \pm 0.004$ | $0.431 \pm 0.017$ | $0.426 \pm 0.021$ | $0.441 \pm 0.013$ |
| $\gamma = 10^{1}$ | $0.421 \pm 0.016$ | $0.427 \pm 0.020$ | $0.436 \pm 0.020$ | $0.445 \pm 0.016$ |
| $\gamma = 10^{2}$ | $0.427 \pm 0.017$ | $0.427 \pm 0.016$ | $0.436 \pm 0.021$ | $0.432 \pm 0.014$ |
| $\gamma = 10^{3}$ | $0.410 \pm 0.027$ | $0.424 \pm 0.019$ | $0.422 \pm 0.019$ | $0.418 \pm 0.015$ |
| $\gamma = 10^{4}$ | $0.422 \pm 0.018$ | $0.423 \pm 0.015$ | $0.441 \pm 0.010$ | $0.443 \pm 0.016$ |
| **USPS** | | | | |
| $\gamma = 10^{-6}$ | $0.778 \pm 0.019$ | $0.783 \pm 0.016$ | $0.784 \pm 0.012$ | $0.784 \pm 0.012$ |
| $\gamma = 10^{-5}$ | $0.775 \pm 0.016$ | $0.774 \pm 0.017$ | $0.778 \pm 0.010$ | $0.782 \pm 0.016$ |
| $\gamma = 10^{-4}$ | $0.781 \pm 0.010$ | $0.760 \pm 0.021$ | $0.772 \pm 0.013$ | $0.774 \pm 0.021$ |
| $\gamma = 10^{-3}$ | $0.758 \pm 0.012$ | $0.788 \pm 0.014$ | $0.771 \pm 0.011$ | $0.784 \pm 0.011$ |
| $\gamma = 10^{-2}$ | $0.765 \pm 0.012$ | $0.775 \pm 0.024$ | $0.772 \pm 0.021$ | $0.775 \pm 0.011$ |
| $\gamma = 10^{-1}$ | $0.773 \pm 0.011$ | $0.787 \pm 0.013$ | $0.774 \pm 0.011$ | $0.776 \pm 0.018$ |
| $\gamma = 10^{0}$ | $0.778 \pm 0.007$ | $0.772 \pm 0.010$ | $0.774 \pm 0.017$ | $0.768 \pm 0.021$ |
| $\gamma = 10^{1}$ | $0.767 \pm 0.018$ | $0.774 \pm 0.013$ | $0.779 \pm 0.016$ | $0.773 \pm 0.014$ |
| $\gamma = 10^{2}$ | $0.774 \pm 0.014$ | $0.782 \pm 0.013$ | $0.776 \pm 0.018$ | $0.771 \pm 0.021$ |
| $\gamma = 10^{3}$ | $0.774 \pm 0.013$ | $0.774 \pm 0.017$ | $0.775 \pm 0.012$ | $0.763 \pm 0.025$ |
| $\gamma = 10^{4}$ | $0.778 \pm 0.013$ | $0.773 \pm 0.012$ | $0.774 \pm 0.012$ | $0.781 \pm 0.011$ |

**Table 2.** Results *(mIoUs)* associated with the experiments on SYNTHIA dataset. The *first* column indicate the training set. The *second* column indicate the method used: Empirical Risk Minimization *(ERM)* and our method *(Ours)* with $K = 1$ and $\gamma = 1.0$. Remaining columns indicate the test set.

| | | New York-like City | | | | | Old European Town | | | | |
|---|---|---|---|---|---|---|---|---|---|---|---|
| | | Dawn | Fog | Night | Spring | Winter | Dawn | Fog | Night | Spring | Winter |
| | *ERM* | 18.9 | 14.7 | 10.7 | 14.5 | 13.4 | 22.0 | 20.8 | 14.5 | 18.6 | 15.3 |
| Highway/Dawn | *Ours* | **24.0** | **17.0** | **19.1** | **22.9** | **20.2** | **27.6** | **25.0** | **22.4** | **27.1** | **19.0** |
| | *ERM* | 12.6 | 27.8 | 9.0 | 12.9 | 13.4 | 13.6 | 20.7 | 12.1 | 15.1 | 12.7 |
| Highway/Fog | *Ours* | **17.4** | **28.4** | **11.0** | **18.4** | **18.4** | **18.5** | **27.5** | **16.4** | **22.0** | **19.0** |
| | *ERM* | 13.0 | 7.7 | 13.9 | 13.2 | 10.9 | 16.6 | 11.5 | 19.0 | 15.7 | 9.9 |
| Highway/Night | *Ours* | **18.5** | **14.5** | **24.8** | **22.9** | **22.0** | **22.2** | **20.1** | **28.1** | **25.5** | **19.1** |
| | *ERM* | 15.2 | 16.0 | 10.8 | 15.8 | 14.8 | 18.8 | 21.2 | 14.7 | 19.2 | 13.9 |
| Highway/Spring | *Ours* | **22.6** | **19.4** | **14.6** | **25.5** | **23.5** | **25.1** | **26.5** | **21.5** | **29.9** | **24.5** |
| | *ERM* | 14.1 | 15.9 | 11.7 | 14.8 | 16.8 | 15.2 | 19.3 | 14.6 | 16.9 | 20.0 |
| Highway/Winter | *Ours* | **16.9** | **17.4** | **12.5** | **21.0** | **24.0** | **17.0** | **20.5** | **14.9** | **23.1** | **26.8** |
| | | Highway | | | | | Old European Town | | | | |
| | | Dawn | Fog | Night | Spring | Winter | Dawn | Fog | Night | Spring | Winter |
| | *ERM* | 19.6 | 19.1 | 13.1 | 18.8 | 15.9 | 27.9 | 23.5 | 16.3 | 21.7 | 17.0 |
| NY.Like C./ Dawn | *Ours* | **22.8** | **22.8** | **17.8** | **21.4** | **18.5** | **31.0** | **25.9** | **22.4** | **26.0** | 22.3 |
| | *ERM* | 12.5 | 15.9 | 9.1 | 11.8 | 10.7 | 24.2 | 26.5 | 17.8 | 21.7 | 16.0 |
| NY.Like C./Fog | *Ours* | **15.4** | **23.1** | **16.3** | **18.7** | **18.2** | 17.3 | **26.4** | 17.5 | **24.3** | 21.6 |
| | *ERM* | 14.9 | 14.7 | 16.3 | 13.5 | 13.1 | 25.4 | 24.7 | 24.4 | 23.3 | 17.0 |
| NY.Like C./Night | *Ours* | **19.4** | **20.2** | **22.1** | **19.7** | **17.3** | 23.3 | 23.9 | **27.2** | **27.2** | **22.1** |
| | *ERM* | **17.1** | **18.0** | **12.8** | **16.3** | **14.8** | **26.6** | **27.0** | **20.4** | **26.3** | 22.5 |
| NY.Like C./Spring | *Ours* | 14.5 | 14.7 | 11.8 | 15.2 | 11.2 | 21.9 | 21.9 | 19.7 | 24.8 | 22.9 |
| | *ERM* | 16.1 | 17.3 | 11.9 | 16.5 | 16.0 | **21.3** | **23.8** | 19.4 | 24.1 | 23.2 |
| NY.Like C./Winter | *Ours* | **18.1** | **18.2** | **15.2** | **17.8** | **17.3** | 21.0 | 21.0 | **19.9** | **25.5** | **25.6** |