[Reviews · NeurIPS 2018]

Reviewer 1



The paper introduces a novel approach for learning models that generalize to new target distributions different from the training one. This is especially interesting for settings where target distributions are unknown, i.e. there is no available data for them (labelled or unlabelled). The paper follows an adversarial training approach similar to (Sinha et al., 2018), i.e. a min-max objective which model parameters minimize an expected loss, where the expectation is taken over the furthest data distributions that are at most ρ distance away from training distribution P0. Crucially, the paper defines a Wasserstein distance between distributions in the semantic space, and proposes an efficient iterative algorithm for optimizing the loss. Experimental evaluation is performed in both digit classification and scene segmentation settings, where the model is trained on one dataset (e.g. MNIST in classification) and tested on multiple different datasets (e.g. SVHN and MNIST-M in classification). Strengths The approach is novel and very interesting The paper is very well written Experimental results are promising, especially the digit classification setting Weakness It is not clear if the semantic Wasserstein distance would be very informative, especially early in training, since representations won’t necessarily be very informative. I would like the authors to comment on this and on any practical issues in training. Results on scene segmentation are not as successful as in digit classification In conclusion, I think this is a strong paper both theoretically and practically which makes a worthy contribution to NIPS.

Reviewer 2



The paper attacks a novel problem: one-shot domain generalization where given samples from a single domain one requires robustness to unknown domain covariate shift. The problem is extremely hard and the related works claim that no other work exists attacking the same problem. The paper incrementally builds up on a recently proposed method to defend against adversarial attacks [33]. In fact, the work uses the formulation of [33] and repurposes the procedure for one-shot domain generalization. The original additions are 3-fold: 1) the closeness constraint is changed from pixel-space to feature-space 2) an ensemble of models are trained with different neighborhood thresholds 3) a new theoretical motivation is provided. It tests the method on the transfer between digit datasets as well as road datasets. The results show that the proposed training work better when tested on unseen domains compared to standard training. The main issue is that, since no prior works exist to compare against, it needs more comprehensive experimental design to show the real benefits of the method. The experimental setup is currently very limited. For instance, I would like to see how are the benefits of using the new method related to the capacity of the model and how regularized it is (by design and by choice of hyperparameters). Would we see the same benefits if the model is highly regularized already? What is the benefit of the constraint in the feature space instead of pixel space? A transition between using different layers would probably give some insight. Some analysis on the samples which are only correctly classified using the proposed method will be helpful. At the same time, an analysis of the samples that are added to the dataset during the training procedure can be insightful. These two sets can be further studied together. All in all, the originality of the paper is lacking, the experimental setup is not convincing, and there are not much insights given by the paper into the novel problem. So, I think, the paper is not ready for publication at NIPS. ------------------------------------------ I read the authors' rebuttal and other reviewers comments. The rebuttal partially mitigates my general concerns with the paper: 1) regularization and 2) general insights. On the other hand, I can see the other reviewers' main positive point that from the theoretical point of view it is a simple and clear work on top of the existing ones. So, I increase my rating slightly but still lean towards the rejection of the paper. I think the paper will become significantly stronger if along the lines of the rebuttal further and more comprehensive studies are pursued.

Reviewer 3



I believe this is an extension of a ICLR'18 work, which advances adversarial training for increasing the robustness in modeling data distributions, so that adapting the learned models to unseen domains can be realized. Overall, this is a solid work with strong theoretical supports. However, given this is an extension of the aforementioned work, I do have some concerns about the experiments. While it is very promising to be able to adapt the learning model to unseen domains without observing target-domain training data, some experimental results did not fully support the proposed method. For example, when training on MNIST to adapt to USPS, the performance was actually worse than that of baseline approaches (i.e., no adaptation). The authors did no provide results on the original source domain data, which makes me wonder if the proposed method would fail in describing/recognizing source-domain data (or data with similar distributions/properties).